# Probing Information Flow in Vision Transformers Through Controlled Attention Perturbation

**Thanh Do**
Stony Brook University

**Abe Leite**
Stony Brook University

## Abstract

We apply identical attention sparsity to three vision transformer tasks and find order-of-magnitude differences in sensitivity: at 75% sparsity, CLIP retrieval degrades 2%, classification degrades 7%, while diffusion generation degrades 274%. To systematically probe this, we design three masking strategies with distinct graph-theoretic properties (small-world, preferential attachment, hub-spoke) and measure degradation across density levels. Ablating small-world masks reveals that spatial locality, not long-range shortcuts, drives performance preservation, with local-only connectivity outperforming random-only by 7.6×. We hypothesize that diffusion's sensitivity arises from error accumulation across 250 sequential denoising steps, where each disruption compounds through subsequent iterations. These findings demonstrate how controlled perturbation can reveal task-dependent differences in transformer information flow that static analysis would miss.

## 1 Introduction

Vision transformers (Dosovitskiy et al., 2021) achieve state-of-the-art performance across diverse tasks by allowing each image patch to attend to all others through self-attention. This creates a dense $N \times N$ connectivity structure where information can flow directly between any pair of tokens. But how much of this connectivity is actually necessary? The quadratic cost of full attention has motivated extensive work on sparse alternatives, yet we lack understanding of which sparsity patterns preserve task performance and why.

We address this through controlled perturbation experiments. Rather than analyzing static attention patterns, we systematically disrupt them using masking strategies with known graph-theoretic properties (Watts & Strogatz, 1998; Barabási & Albert, 1999), then measure how task performance degrades. This intervention approach reveals which connectivity patterns matter for which tasks.

Our experiments yield a surprising finding: task-dependent sensitivity varies by orders of magnitude under identical perturbations. At 75% attention sparsity, CLIP retrieval degrades only 2%, classification degrades 7%, and diffusion generation degrades 274%. While the models differ in depth and width (ViT-L/16, CLIP ViT-B/32, DiT-XL/2), all use standard self-attention mechanisms, suggesting that the computational structure of the task itself determines robustness to attention disruption.

We hypothesize that diffusion's sensitivity arises from error accumulation across its 250 sequential denoising steps. Each step's output becomes the next step's input, so attention disruptions compound multiplicatively rather than being absorbed. Classification and retrieval, by contrast, perform single forward passes where local errors remain local. This has practical implications: sparse attention methods developed for discriminative tasks may fail when applied to iterative generative processes.

## 2 Related Work

The quadratic cost of self-attention has driven sparse alternatives. BigBird (Zaheer et al., 2020) combines local windows, random edges, and global tokens; Longformer (Beltagy et al., 2020) uses

sliding windows with global attention. These achieve efficiency through *designed* sparsity, trained from scratch. We instead probe *pretrained* models through post-hoc masking, revealing which connectivity patterns they actually require. Our masking strategies draw on classical network models: small-world networks (Watts & Strogatz, 1998), preferential attachment (Barabási & Albert, 1999), and hub-spoke topologies (O'Kelly, 1986).

Recent work demonstrates that sparse attention can be highly effective when incorporated during training. DiTFastAttn (Yuan et al., 2024) achieved 76–88% FLOP reduction through window attention; PiT (Wu et al., 2025) found 99% of tokens have attention distances under 6 pixels. The Cannistraci group shows 1–5% connectivity can match full performance when trained appropriately (Zhang et al., 2024). These successes with *training-time* sparsity motivate a complementary question: what connectivity do *pretrained* models actually require? We address this through post-hoc masking, revealing which attention patterns matter after training has shaped the model's information pathways.

## 3 METHODS

### 3.1 ATTENTION AS GRAPH STRUCTURE

Self-attention creates an implicit graph where tokens are nodes and attention weights define directed edges. For input $\mathbf{X} \in \mathbb{R}^{N \times D}$, attention computes:

$$\mathbf{A} = \text{softmax}\left(\frac{\mathbf{Q}\mathbf{K}^\top}{\sqrt{d_k}}\right), \quad \text{where } \mathbf{Q} = \mathbf{X}\mathbf{W}_Q, \ \mathbf{K} = \mathbf{X}\mathbf{W}_K \tag{1}$$

The attention matrix $\mathbf{A} \in \mathbb{R}^{N \times N}$ can be interpreted as a weighted adjacency matrix where $A_{ij}$ indicates information flow from token $j$ to token $i$. We probe this structure through controlled masking: computing $\mathbf{A}' = \mathbf{A} \odot \mathbf{M}$ where $\mathbf{M}$ is a binary mask, then renormalizing each row by its sum to maintain valid probability distributions (i.e., $A'_{ij} \leftarrow A'_{ij} / \sum_k A'_{ik}$). If a row sums to zero (isolated token), we set it to uniform attention over allowed connections. The same mask $\mathbf{M}$ is applied identically across all layers and all attention heads within each model.

### 3.2 MASKING STRATEGIES

We design three masking strategies with distinct topological properties.

**Small-World Masking.** This strategy is inspired by the Watts-Strogatz model (Watts & Strogatz, 1998), which combines local clustering with random long-range shortcuts. Each token $i$ at grid position $(r_i, c_i)$ connects to all tokens within Manhattan distance $R$, defined as $\mathcal{N}_{\text{local}}(i) = \{j : |r_i - r_j| + |c_i - c_j| \leq R\}$. Additionally, each token samples $S$ random shortcuts independently from outside its local neighborhood. Shortcuts are directed and asymmetric, meaning if $i \to j$ is sampled, $j \to i$ is not automatically included. Self-connections are always permitted. We denote configurations as R$x$S$y$ (e.g., R4S32 indicates radius 4 with 32 shortcuts).

**Preferential Attachment Masking.** This strategy creates scale-free networks following the Barabási-Albert model (Barabási & Albert, 1999). Starting from 2 initial connections per token, we add edges iteratively by sampling targets with probability $P(j \mid i) \propto (\deg_{\text{in}}(j) + 1)^\alpha$ until reaching target density $D$, defined as the fraction of $N^2$ possible edges. Higher values of $\alpha$ create stronger hub concentration, testing whether information can route through a few high-degree tokens.

**Hub-Spoke Masking.** This strategy is inspired by hub-and-spoke network architectures common in transportation and communication systems (O'Kelly, 1986). We designate $K$ hub tokens selected uniformly at random, which form a complete subgraph among themselves. Non-hub tokens connect only to their nearest hubs by Manhattan distance. This tests extreme centralization: whether a small number of hubs can serve as sufficient information bottlenecks.

### 3.3 TASKS, MODELS, AND METRICS

We select three tasks with fundamentally different computational structures, specifically to vary *path dependency*: the degree to which outputs depend on sequential accumulation of intermediate computations.

Table 1: Performance across tasks and masking strategies at 25% and 50% density. For DiT, CMMD values are shown (lower is better); percentage changes indicate degradation from baseline. At 25% density, CLIP retains 93–97% of baseline recall while DiT CMMD increases 274–638%.

| | | Masking Strategy | | |
|---|---|---|---|---|
| Task (Baseline) | Density | Small-World | Pref. Attach. | Hub-Spoke |
| CLIP R@1 | 25% | 0.79 (−2%) | 0.75 (−8%) | 0.64 (−21%) |
| (0.81) | 50% | 0.81 (−0%) | 0.80 (−1%) | 0.74 (−9%) |
| Classification | 25% | 78.9 (−7%) | 62.6 (−26%) | 69.2 (−19%) |
| Acc. (85.0%) | 50% | 84.2 (−1%) | 82.7 (−3%) | 82.5 (−3%) |
| DiT CMMD | 25% | 1.98 (+274%) | 3.91 (+638%) | 3.61 (+581%) |
| (0.53) | 50% | 1.21 (+128%) | 2.81 (+430%) | 3.02 (+470%) |

**Classification.** We use ViT-L/16 (Dosovitskiy et al., 2021) on ImageNet (Deng et al., 2009), which performs a single forward pass to produce a 1000-way decision. Errors at any layer can be compensated by redundant pathways, and the final decision aggregates evidence without temporal dependencies. This task has low path dependency. We measure Top-1 accuracy on the ImageNet validation set (50K images).

**Vision-Language Retrieval.** We use CLIP ViT-B/32 (Radford et al., 2021) on COCO (Lin et al., 2014), which performs contrastive matching between image and text embeddings. Small perturbations may shift embeddings but preserve relative similarities if the representation is robust. This task has low-to-medium path dependency. We measure Mean Recall@1 (average of image-to-text and text-to-image retrieval) on COCO validation (5K images).

**Diffusion Generation.** We use DiT-XL/2 (Peebles & Xie, 2023), which performs iterative denoising using the canonical DDPM schedule (Ho et al., 2020) over 250 timesteps. Each step $x_{t-1} = f(x_t, t)$ takes the previous output as input, so errors compound: attention disruption at step $t$ corrupts the input for all subsequent steps. This task has high path dependency. We generate 1000 images per configuration using class-conditional generation on ImageNet classes.

For diffusion, we use CLIP-based mean embedding distance (CMMD) (Jayasumana et al., 2024) rather than FID (Heusel et al., 2017):

$$\text{CMMD} = \left\| \frac{1}{n} \sum_{i=1}^{n} \phi(g_i) - \frac{1}{m} \sum_{j=1}^{m} \phi(r_j) \right\|_2 \tag{2}$$

where $\phi$ is the CLIP image encoder, $\{g_i\}$ are generated images, and $\{r_j\}$ are ImageNet reference images. This metric measures the L2 distance between mean CLIP embeddings of generated and reference distributions. We prefer CMMD over FID for two reasons demonstrated by Jayasumana et al. (2024): first, CMMD provides stable estimates with fewer samples ($\sim$1000 vs $\sim$50000 for FID), which is critical when evaluating many masking configurations; second, CLIP embeddings better capture semantic content than Inception features used by FID. Lower values indicate higher quality.

## 4 RESULTS

### 4.1 TASK-DEPENDENT SENSITIVITY

Table 1 reveals striking task-dependent sensitivity. At 25% density, CLIP degrades only 2–21% depending on mask type, classification degrades 7–26%, while DiT degrades 274–638% (over an order of magnitude more sensitive under identical sparsity). At 50% density, CLIP and classification are near-baseline ($\leq$9% degradation), but DiT still suffers 128–470% quality loss. Across both densities, small-world masking consistently outperforms the alternatives, achieving the lowest degradation on every task.

Figure 1 visualizes the degradation curves. CLIP's curve remains nearly flat (97% of baseline at 15% density), classification shows a sharp cliff below 10% density, and diffusion degrades continuously

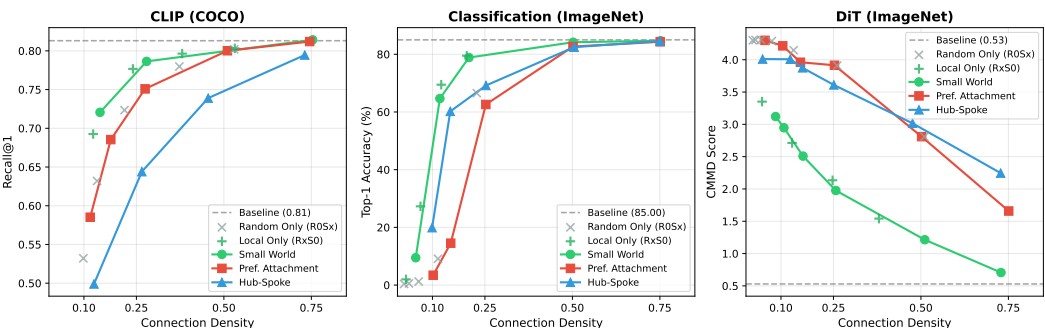

Figure 1: Degradation curves across three tasks under three masking strategies. Dashed lines indicate baselines; scatter points show local-only (R$x$S0: radius $x$, zero shortcuts) and random-only (R0S$x$: zero radius, $x$ shortcuts) ablations. **Left:** CLIP (COCO, baseline R@1=0.81) is highly robust; small-world masking retains 97% of baseline at 15% density. **Center:** Classification (ImageNet, baseline 85.0%) shows a sharp cliff below 10% density. Local-only dramatically outperforms random-only at matched density (69.5% vs. 9.1% at ∼13%). **Right:** DiT (baseline CMMD=0.53, lower is better) degrades rapidly; even at 50% density, small-world CMMD is 1.21 (+128%). Hub-spoke and preferential attachment perform substantially worse than small-world across all tasks and densities.

with no safe operating region. At matched density (∼25%), small-world masking achieves CMMD 1.98 versus hub-spoke's 3.61 and preferential attachment's 3.91, suggesting distributed local-plus-shortcut structure preserves information flow better than centralized or scale-free routing.

## 5 DISCUSSION

**Error accumulation in iterative processes.** We hypothesize diffusion's sensitivity arises from multiplicative error accumulation. Each denoising step $x_{t-1} = f(x_t, t)$ operates on the previous output; disruptions at step $t$ corrupt all 249 subsequent steps. Classification and CLIP perform single passes where errors cannot compound temporally.

**Local structure dominates random shortcuts under post-hoc masking.** Ablating the small-world mask into its components reveals a striking asymmetry: local-only connectivity (radius $R$, zero shortcuts) nearly matches full small-world performance, while random-only connectivity (zero radius, $S$ shortcuts) degrades catastrophically. At ∼13% density on classification, local-only (R6S0) achieves 69.5% accuracy versus random-only (R0S64) at 9.1%, a 7.6× gap. On DiT at ∼14% density, local-only CMMD is 2.71 versus random-only's 4.15. Adding shortcuts to local connectivity provides marginal gains: at 20% density on classification, local-only R8S0 achieves 79.5% versus small-world R5S64 at 78.9%. This suggests that *when disrupting pretrained models*, preserving spatially local attention pathways matters far more than preserving random long-range connections. The practical implication is that simple windowed attention may be nearly as effective as more complex small-world designs for post-hoc sparsification.

**Reconciling with efficient diffusion work.** DiTFastAttn and PiT achieve sparse diffusion through architectural design during training. Our post-hoc masking fails because it disrupts learned pathways. The distinction matters: sparse attention is achievable for diffusion but must be incorporated during training, not retrofitted afterward.

**Limitations.** We test one diffusion architecture (DiT-XL/2). The error accumulation hypothesis requires direct measurement of per-step propagation to confirm. Alternative explanations (higher bandwidth requirements, global coherence sensitivity) remain plausible.

## 6 CONCLUSION

Through controlled perturbation experiments, we found that vision transformer tasks have dramatically different sensitivity to attention disruption: CLIP remains robust while diffusion fails catas-

trophically under identical masking. Ablating small-world masks further reveals that spatial locality, not long-range shortcuts, is the primary driver of performance preservation under sparsity. These findings emerge from systematic intervention rather than observation, demonstrating how experimental methods can reveal task-dependent structure that static analysis would miss.

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
