# OpenReview forum: "PROBING INFORMATION FLOW IN VISION TRANSFORMERS THROUGH CONTROLLED ATTENTION PERTURBATION"
_ICLR.cc/2026/Workshop/Sci4DL — Sci4DL 2026_

### Official Review · Reviewer_aCmA · 2026-02-08

**Fit:** 3
**Significance:** 2
**Confidence:** 2

**Summary:**

The authors find that sparsifying the attention mechanism (via masking) in vision transformers yields significantly different sensitivities in terms of task performance. Namely, in some tasks, sparse attention barely degrades performance (e.g., CLIP retrieval), whereas in others it significantly degrades performance (e.g., diffusion generation). The authors provide an interesting hypothesis for such a major difference, suggesting that sparse attention methods developed for discriminative tasks may fail when applied to generative processes (in which small errors accumulate iteratively).

**Strengths:**

The manuscript is overall clear and written in a manner that is "to the point." The problem statement, results, and proposed methodologies are written clearly from the beginning of the paper. The results themselves are interesting and could have important implications for future designs of vision transformer attention mechanisms (at least in terms of their sparsity, suggesting task-dependent design).

**Suggestions:**

Could the results guide the design of some form of universal attention mechanism that is task-dependent? Namely, is there a way to design an attention mechanism that automatically "sparisifies" to the correct level, depending on the task the model is trying to solve?

---

### Official Review · Reviewer_ozMG · 2026-02-14

**Fit:** 2
**Significance:** 2
**Confidence:** 2

**Summary:**

The authors apply the same binary sparsity mask (shared across all layers and heads) to three pretrained  tasks classification , retrieval and  generation, and measure performance degradation as attention density decreases.
They design three mask families with different graph properties: small-world (local neighborhood + random shortcuts), preferential attachment (hub-heavy scale-free), and hub-spoke (explicit hubs).
Across densities, they observe orders-of-magnitude task sensitivity differences under identical sparsity: discriminative tasks (classification/retrieval) remain relatively robust at moderate sparsity, while diffusion quality degrades sharply.

**Strengths:**

The paper’s experimental design is especially clean: by applying identical attention masks across different tasks, it isolates task-dependent sensitivity in a way that is unusually direct and difficult to dismiss. Moreover, the choice of tasks is conceptually grounded in “path dependency”—contrasting single-pass objectives (e.g., classification/retrieval) with iterative refinement (diffusion)—which offers a coherent framework for interpreting why sparsification affects these settings so differently.

**Suggestions:**

A key limitation is that the choice of masking families (small-world, preferential attachment, hub-spoke) is only loosely motivated, and the paper’s central explanatory claims—particularly why locality dominates robustness and why diffusion is dramatically more sensitive—remain largely hypothesized rather than causally validated (e.g., without direct per-step error-propagation analysis). This weakens the strength of the mechanistic conclusions and may limit the paper’s broader impact.

---

### Meta-Review · Area_Chair_SCsa · 2026-03-02

**Recommendation:** Accept

**Metareview:**

In this work, the authors study how sparsifying the attention mechanism (via masking) in vision transformers yields significantly different sensitivities in terms of task performance. I find the work insightful and relevant to the workshop.

I recommend an accept.

---

### Decision · Program_Chairs · 2026-03-02

Accept